# The Natural Consequences of Land Use Change on Transformation and Vegetation Development in the Upper Odra Floodplain

Agnieszka Czajka, Oimahmad Rahmonov * and Bartłomiej Szypuła

Institute of Earth Sciences, Faculty of Natural Sciences, University of Silesia in Katowice, Będzińska 60, 41-200 Sosnowiec, Poland; agnieszka.czajka@us.edu.pl (A.C.); bartlomiej.szypula@us.edu.pl (B.S.)
* Correspondence: oimahmad.rahmonov@us.edu.pl

**Abstract:** River channels are regulated in various ways and the fertile soils of valleys are occupied for agricultural purposes, accompanied by human settlements. In many places on the floodplains, gravel or sand is mined and former pits fill with water. The consequences are changes in water relations, changes in land use and land cover. Natural riparian ecosystems gradually disappear. In addition, river valleys are susceptible places for the spread of invasive plant species. In the section of the Upper Odra Valley discussed in this article, all of the aforementioned factors have played roles in shaping modern habitats. The present study shows the impact of human-induced changes on the transformation of the plant cover of the Upper Odra Floodplain. In designated transects, we studied land use changes from 1910 to the present day and examined plant species diversity. The results show that the more heavily transformed floodplain adjacent to the channelized channel has a higher level of species diversity than agricultural areas located along a section of the river with a natural channel course. Most of the river valleys are colonized by geographically invasive alien species, such as *Reynoutria japonica*, *Reynoutria sachalenesis* and *Impatiens glandulifera*, which have contributed to the fact that all of the species typical of the ash, poplar and willow riparian forests characteristic of this habitat type have retreated, which is the main reason for the very low biodiversity.

**Keywords:** floodplain land cover changes; floodplain biodiversity; invasive species; floodplain ecosystems





## 1. Introduction

River valley ecosystems are some of the most vulnerable areas to human transformation on Earth. The high biodiversity of the floodplains of natural rivers is closely linked to the constant remodeling of relief that rejuvenates riparian habitats [1] but, nowadays, the functioning of the natural environment increasingly depends on human activity and its intensity [2–7]. Man influences the environment both in the use of natural resources and in taking various types of conservation measures [8] with the aim of maintaining ecological balance. Human economic activities and the development of hitherto unused land lead to the disintegration of both aquatic and terrestrial ecosystems worldwide [2,9]. In both historical and modern aspects, river valleys have been the most developed for both settlement and agricultural purposes.

River valleys, with their sensitive ecosystems, are susceptible to uncontrolled external factors, which may lead to their complete transformation both in terms of water and habitat relations. Thus, they may lose their ecosystem services [10]. In many regions of Europe, river valleys are long ecological corridors that are protected as Natura 2000-designated areas. One of the effects of human activity is the creation of completely new habitats on river floodplains. These areas are shaped on large spatial and temporal scales [11]. Transformed areas cover about 6% of the land area [12]. It is estimated that human activity covers about 1% of the area [13], so floodplains are an important element of the landscape

of many places in the world and constitute habitats for various systematic groups of both animals and plants, which are often subject to anthropopressure.

The Odra River is regulated along almost its entire length. As a result of the construction of various hydrotechnical structures, the natural dynamics of the riverbed have been disturbed and strongly varied [14]. Most of the hydrotechnical works, including straightening the riverbed with concreting, the artificial construction of embankments and floodplain meliorations, have resulted in a complete transformation of the alluvial willow, alder, ash and other wetland vegetation ecosystems and their replacement by other anthropogenic plant associations [15–17]. The most invasive alien species that form multi-layered communities along the riverbed and floodplain of the Odra River are the Asian species *Reynoutria japonica* and *R. sachalinensis* [18], which are geographically alien to this region and willingly occupy similar habitats. In addition, significant amounts of municipal and industrial wastewater are discharged into the river, causing pollution. The consequence of ecosystem degradation is the reduction or even complete loss of their ecosystem functions. These changes became more visible both in plant communities and in their species composition in various sections of the flood zones. Vegetation is one of the most sensitive elements of the natural environment, which can be treated as an indicator of changes in both biotic and abiotic conditions of the river ecosystem.

The anthropogenic disturbance of riparian ecosystems leads to a reduction in biodiversity and synanthropization of the existing vegetation, and also creates conditions for the entry of alien or unusual species for a given habitat [19,20]. This often manifests itself in the increase in the number of anthropogenic communities, the disappearance of original species combinations, the creation of new heterogeneous species combinations, the creation of species-poor and less diverse communities and even the artificial planting of trees through forest reclamation [7,18,21–23].

Floodplain vegetation provides many ecosystem services, such as protection against soil erosion, water retention, transport of matter, self-purification of rivers and habitats for many organisms. The occurrence of breeding communities (especially willow ones) in the form of isolated islands in the flood zone used for agricultural purposes indicates their earlier occurrence throughout the area. Thus, they are the biological memory of this very important forest ecosystem in river valleys. The transformation of river valleys as a result of hydrotechnical works and disturbance of water relations may lead to the complete and irreversible destruction of riverside ecosystems together with biocenotic and social functions [24]. The impact of industrial and agricultural human activities on the transformation of river valleys has been documented in many regions of the world, most often manifested within urban boundaries in the form of river channel regulation [25–30]. Therefore, the aim of this study was to determine land use changes in the river floodplain (1) and their impact on vegetation diversity in naturally meandering and channelized sections (2) against the background of environmental conditions.

## 2. Materials and Methods

### 2.1. Study Area

The Odra River is the second largest river in Poland with a total length of 854.3 km, 742 km of which is within Poland. The area of the Odra River drainage basin is 118,861 km$^2$ and it drains almost 90% of Poland's territory.

The Odra flows north from the Czech Republic and enters Poland through the Moravian Gate, which is a depression between the Carpathian Mountains to the east and the Sudety Mountains to the west. Here, on the Polish-Czech borderland, a 9 km-long section of the Odra River Valley was chosen to study the relationship between floodplain use and vegetation diversity. Due to the different nature of the river channel, this section was further divided into two subsections: channelized and naturally meandering (Figure 1). The catchment area of the studied section of the Upper Odra River is ca. 7500 km$^2$ and the average discharge is about 58 m$^3 \cdot$s$^{-1}$. The average annual air temperature is 8.5 °C,

(18.5 °C in July and −2.5 °C in January). The growing season is also among the longest in Poland, at 225 days. The amount of precipitation per year is about 750 mm [31].

The upper Odra River flows partially through mining areas, which resulted in the need to regulate the river for shipping purposes to transport coal. The regulation works in the studied section began in 1859 and were completed in 1822 [32,33]. The regulation work included reducing the sinuosity of the channel by cutting off meander bends, reducing the width of the channel by the construction of groins and reducing the width of the active floodplain using dikes. As a result of the shortening of the channel, the sinuosity of the channel in the studied section decreased by 36% and the channel slope increased by 37%. The average width of the current channel varies from 30 to 50 m [33].

At the same time, the section of the Odra River upstream from the Olza confluence remains unregulated since it constitutes the border between the Czech Republic and Poland. It is currently the last preserved meandering section of the Odra Channel in Poland. The length of this section is 7 km and the channel geometry changes observed here are the result of natural processes. The natural cutting off of two meanders, one in 1966 and one in 1997, has shortened the course of the river by 1200 m and increased the slope from 0.7 to 1.6% [34].

*2.2. Cartographic Works*

To evaluate long-term land cover changes, a comparative analysis of maps from different periods was conducted. Based on historical and contemporary maps, how the land use and land cover of the Odra Valley changed between 1910 and 2022 was traced.

Archival and recent materials were used to examine long-term land use changes in the studied section of the Odra Valley. The archival materials consisted of the following topographic maps: the German *Topographische Karte Mestischblatt 1:25,000*, sheets: Haatsch (1902, 1912, 1941, 1944) and Gr. Gorschuetz (1912, 1941); and the Polish: *WIG Maps 1:25,000*, sheets: P49-S26 (1939) and P49-S27 (1938); *Topographic Map 1:10,000*, sheets: 494.212 (1984), 494.214 (1984), 540.224P (1984) and 540.222 (1984). As for modern materials, the following were used: vector database of topographic objects (*BDOT10k, 2022*), which corresponds in detail to a topographic map at a scale of 1:10,000; and an orthophotomap of 2021 with a resolution of 0.25 m in real RGB colors, sheets: M-34-73-B-a-1-4, M-34-73-B-a-2-3, M-34-73-B-a-2-4, M-34-73-B-a-3-2, M-34-73-B-a-3-2, M-34-73-B-a-4-1, M-34-73-B-a-4-2, M-34-73-B-a-4-3 and M-34-73-B-a-4-4. In addition, altimetric survey data in LAS format derived from airborne laser scanning (ALS) with a density of min. 4 points/m$^2$ and an average height error of 0.15 m was used. A digital elevation model with a resolution of $1 \times 1$ m was created from these height data.

The cartographic work consisted of several steps: (1) compilation of cartographic, imagery and elevation materials from web services (Mapster, Geoportal); (2) preparation of archival materials for working in the GIS environment: (a) georeferencing of historical topographic maps (ie. from 1902–1944); both Polish and German maps have been calibrated to the PUWG-1992 system (EPSG:2180), which is the applicable system for topographic studies at these scales; and (b) vectorization of information from archival topographic maps and orthophotos; (3) proper cartographic analysis: determination of basic land use classes, determination of the course of the river channel before regulation, calculations related to the areas occupied by the various land use classes, slope and winding of the river, etc., making transect profiles.

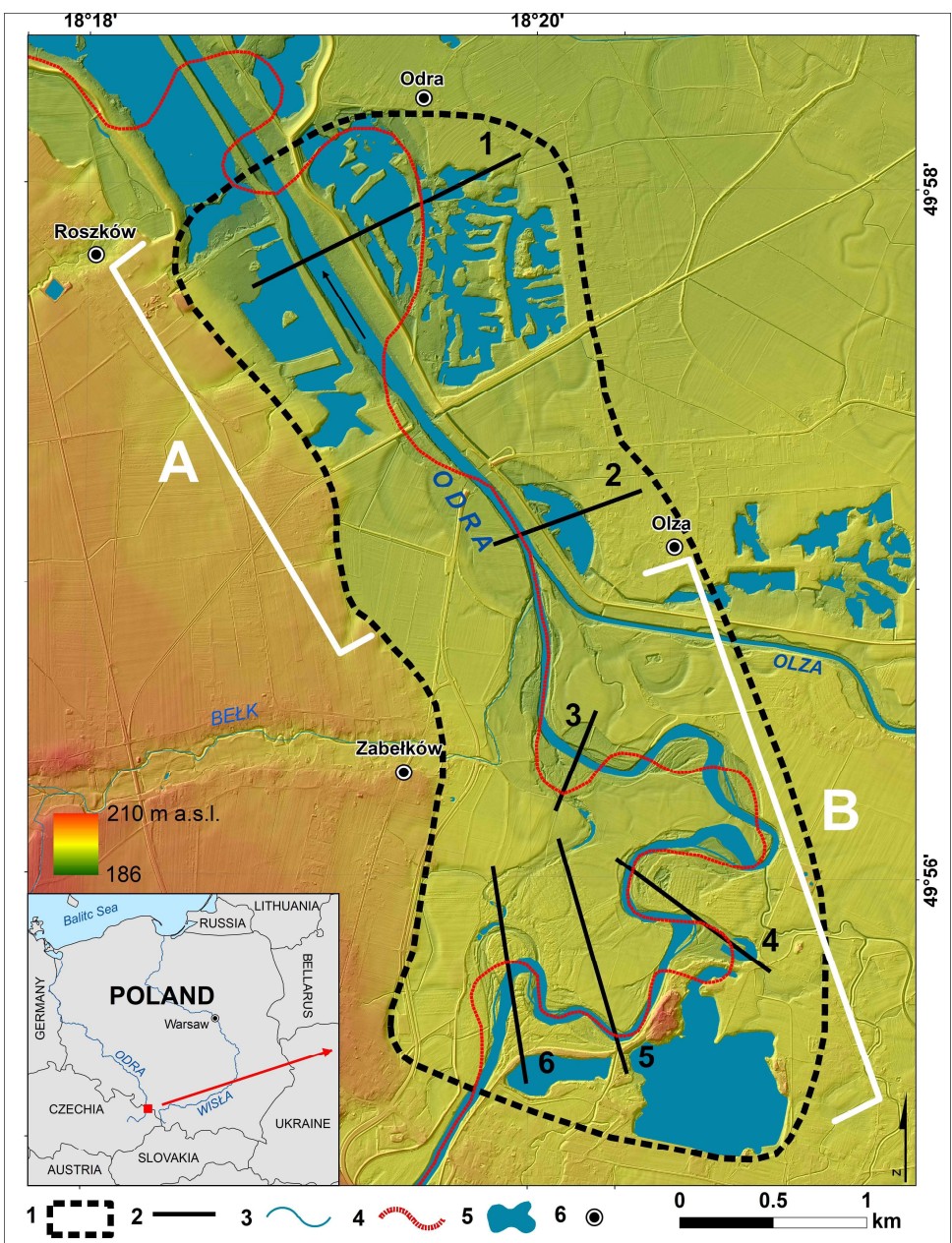

**Figure 1.** Localization of study area: (A) transects along the channelized section of river; (B) naturally meandering section of Odra River; (1) study area boundary; (2) transect lines; (3) present channel course; (4) former channel course; (5) water bodies; and (6) residential areas.

## 2.3. Vegetation Research

A survey of the species composition of vegetation on the Odra floodplain took place in June 2023. In order to recognize the diversity and distribution of vegetation in the different zones of the Odra Valley, six survey transects were selected (Figure 1). The main criterion for the selection of the transects was the degree of naturalness of the river channel flowing through the given section of the valley. The entire valley section is located in a pre-mountainous area that is poorly differentiated geomorphologically. The current differentiation is mainly due to the construction of dikes and the presence of numerous active and inactive gravel pits. The geometry of the Odra Channel itself has been most influenced by regulation (Figure 1) and the natural cutting of meanders in the upper part of the studied section. Transects were located across the river valley (Figure 1) in areas of

varying terrain (regardless of their natural or anthropogenic surroundings). The selection of transects was based on a spatial-temporal analysis, which showed that natural changes such as changes in the extent of vegetation and, consequently, the appearance of invasive species, were mainly due to changes in land use. The length of the transects depended on the land use zone, and their width was always 20 m. The transects were set through different zones, such as the riparian zone, floodplain, dikes and oxbow lakes. Transects 3 and 5 passed only through the floodplain. Plant species were identified based on the vascular plant identification key [35]. Plant names are given according to *Flowering Plants and Pteridophytes of Poland* [36].

## 3. Results

### 3.1. Changes in Land Use and Land Cover between 1910 and 2022

One of the effects of the regulation of the Odra River Channel in the studied section is a drastic change in the basic parameters of the river channel in the channelized section (Table 1) and the initiation of a different economic use of the floodplain than in the meandering section. The current mosaic of habitats on the Odra River floodplain along both studied sections is the result of a variety of human activity (Figure 3). After the Second World War, along the regulated section of the Odra River, gravel mines started to operate in the cut-off meanders. As a result of the intensive mining, numerous gravel pits have formed in the valley floor. On the right bank, after mining ended, the pits filled with water and the reservoirs thus created began to be used for recreational purposes. Gravel mines on the left bank of the Odra were established in the 1980s and are still in operation. These practices have led to a 15.2% increase in the area of stagnant water compared to the state in 1910. In the same section, the active flood zone has been narrowed by dikes to about 200 m. As a result, the range of the flooding zone was limited to the interembankment zone, differentiating habitat conditions for vegetation.

**Table 1.** Regulation-induced changes in the slope and curvature of the Odra riverbed in the studied section.

| Parameter | Entire Section | | Channelized Section | | Meandering Section | |
|---|---|---|---|---|---|---|
| | 1910 | 2022 | 1910 | 2022 | 1910 | 2022 |
| Gradient [°/₀₀] | 0.80 | 0.72 | 0.08 | 0.41 | 1.23 | 0.86 |
| Sinuosity | 2.18 | 1.66 | 1.61 | 1.00 | 2.20 | 2.03 |

The area adjacent to the channel of the Odra River in its meandering section is mostly occupied by arable land. Only patches of riparian forest are fragmentarily preserved.

#### 3.1.1. Transect 1: Anthropogenically Transformed Part of Valley

During the period under analysis, the area where Transect 1 was set has changed considerably. In 1910, most of the area was occupied by farmland and, on the right bank, there was a water-filled, artificially cut-off meander, the banks of which were overgrown by a narrow strip of forest. The 1940 map shows that this forest was cut down and gravel mining began, enlarging the surface water area. The mining area gradually expanded and the created water bodies were separated by narrow dikes, on which willow forests similar to today's ones probably grew. In the 1980s, gravel mining also began on the left bank of the Odra. The same period also saw an increase in urbanized areas, which in the 21st century also included patches of land between the former gravel pits. These newest built-up areas are recreational cottages. After the end of gravel mining from the valley floor on the right bank of the Odra Valley, mining began on the left bank. The area of the pits has gradually expanded, and thus, the total area of surface water has increased. The narrow strip of forest overgrowing the right bank of the Odra in the past also disappeared. Today, this place is occupied by a meadow (Figure 2A). These dynamic changes in land use have also changed the biotope of this part of the valley.

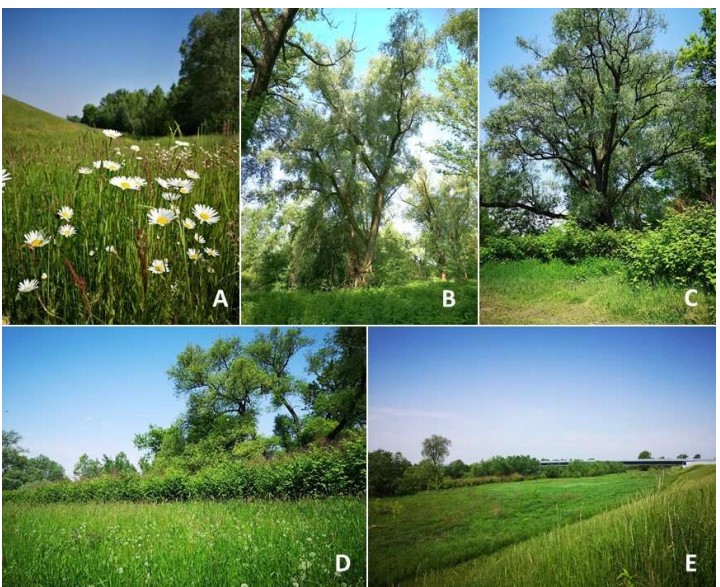

**Figure 2.** Typical vegetation habitats on the floodplain along the studied section of the Odra River (description in text).

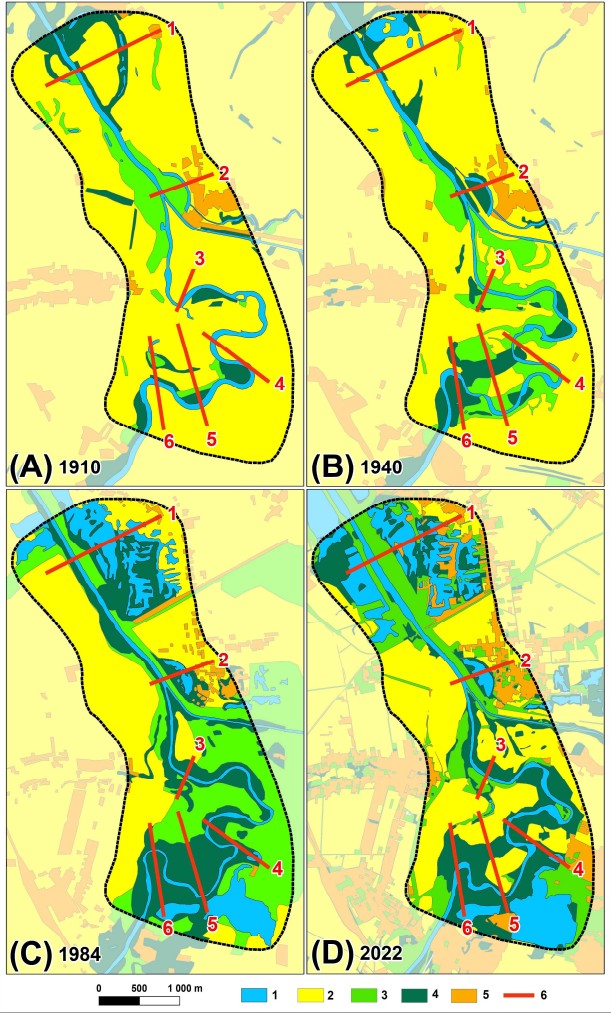

**Figure 3.** Changes of land cover in study site between 1910 and 2022: (1) surface water; (2) arable land; (3) grassland; (4) forest; (5) urban area; and (6) transects.

### 3.1.2. Transect 2: Embankments and Oxbow Lake

The line of Transect 2 crosses a heterogeneous area, where changes in land cover have also occurred. On the left bank of the river, arable land now functions in place of former grassland areas. Along the bank itself, only single trees are growing. On the right bank of the river, the narrow area between the riverbank and the dike is overgrown with grasses and single trees (Figure 2E). On the outer side of the dike, an oxbow lake is located. It has never been used for gravel mining and the adjacent area is overgrown with forest (Figure 4B), but its area is no longer expanding due to its proximity to the built-up area. In the 1940s, the grassland on the left bank of the river was taken up for cultivation. Around the same time, the banks of the oxbow were already covered with forest, the area of which remains unchanged.

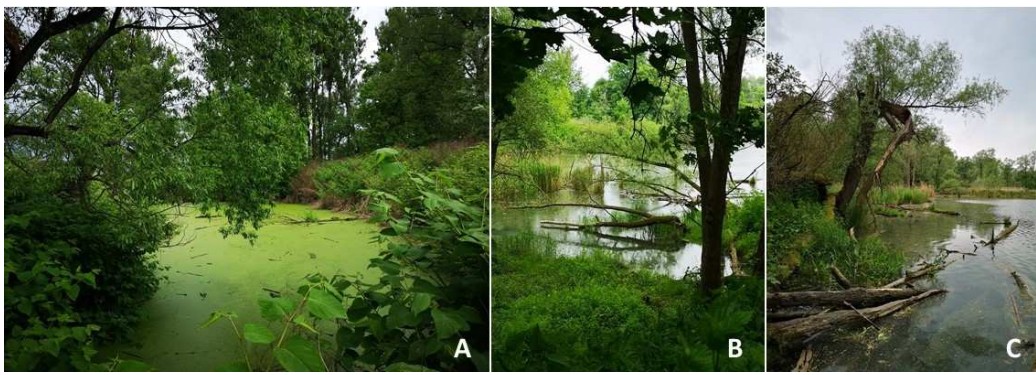

**Figure 4.** Odra oxbow lakes in Transect 6 (**A**), in Transect 4 (**B**) and in Transect 2 (**C**) (description in text).

### 3.1.3. Transect 3: Arable Areas and Wet Meadows

The line of Transect 3 passes through an area continuously used for agriculture, but some changes in land use were also noted here during the period under analysis. At the beginning of the 20th century, the floodplain here was used for crops, but by the middle of the century, the land adjacent to the riverbed on both banks was used as meadows. Over time, the riverbanks became overgrown with trees, forming a willow riparian which directly borders the meadow areas (Figure 2D).

### 3.1.4. Transect 4: Riverside and Oxbow Lake

Transect 4 was delineated in an area where both land use and the course of the riverbed have changed. The area used as arable land has been gradually overgrown with riparian forest. However, this forest is limited to a narrow strip along both banks of the river. On the Czech bank of the Odra in this particular location, this strip is wider than on the Polish bank. In 1966, the meander of the river in the transect area was cut by natural processes, resulting in the formation of an oxbow lake, which still functions today as a reservoir of standing water (Figure 4C). An artificial pit was created near the oxbow lake in the second half of the 20th century. It is currently filled with water, which has contributed to an increase in the total area of surface water within the Odra floodplain in the surveyed section. Part of this artificial reservoir is currently used for recreational purposes and another part is being mined for gravel.

### 3.1.5. Transect 5: Willow Communities on the Former Oxbow

The area through which Transect Line 5 runs has undergone significant changes in terms of land use. At the beginning of the 20th century, most of the land here was farmland. Meadows occupied a small area, as did the riparian forest, which at the time existed in the form of two patches: the larger one overgrew the left bank of the Odra, and the other was a marshy depression, still visible in the relief of the land, where water periodically flows. Since the mid-20th century, this area of forest has gradually expanded and, in the 1980s, it

occupied the maximum area. Today, it is dominated by willow riparian forests (Figure 2B) and arable land.

3.1.6. Transect 6: Riverside Forest and Water Communities

The line of Transect 6 runs through an area occupied today entirely by riparian forest. Its area has increased gradually since the beginning of the 20th century, occupying land previously used for cultivation. A comparison of maps from 1910 to 2022 shows the significant dynamics of the Odra meander in this area. It was finally cut during the 1997 flood and is now an oxbow lake with no fresh water supply (Figure 4A). The area adjacent to this oxbow lake is overgrown with trees.

The study area is a mosaic of arable land, grassland, forests, surface waters and rural built-up areas. The arrangement of the various elements of this puzzle changed over the period 1910–2022 (Table 2). The area of surface waters, grasslands and forests increased. Built-up areas, including summer houses around artificially created water reservoirs, also increased. In contrast, the area of arable land decreased significantly.

**Table 2.** Area and percent shares of land cover classes 1910–2022 in study site.

| Land Use Class | 1910 | | 1940 | | 1984 | | 2022 | |
|---|---|---|---|---|---|---|---|---|
| | Area | | | | | | | |
| | km$^2$ | % | km$^2$ | % | km$^2$ | % | km$^2$ | % |
| Water bodies | 0.09 | 0.9 | 0.19 | 1.9 | 1.22 | 12.3 | 1.60 | 16.1 |
| Arableland | 8.02 | 80.6 | 7.07 | 71.1 | 3.55 | 35.7 | 3.47 | 34.9 |
| Grassland | 0.76 | 7.6 | 1.45 | 14.6 | 2.40 | 24.1 | 1.84 | 18.5 |
| Forests | 0.76 | 7.6 | 0.93 | 9.3 | 2.54 | 25.5 | 2.18 | 21.9 |
| Buildings | 0.32 | 3.2 | 0.31 | 3.1 | 0.24 | 2.4 | 0.86 | 8.6 |
| Total | 9.95 | 100.0 | 9.95 | 100.0 | 9.95 | 100.0 | 9.95 | 100.0 |

*3.2. Vegetation and Floristic Diversity*

3.2.1. Non-Forest Vegetation

On the transects surveyed, the occurrence of vegetation is conditioned by the nature of the habitat. The dikes (irrespective of their size) are overgrown by a community of the class *Molinio-Arrhenatheretea*, order *Arrhenatheretalia elatioris*. The assemblage is composed of *Arrhenatheretum elatioris* with characteristic species such as *Arrhenatherum elatius, Vicia cracca, Phleum pratense, Lotus corniculatus, Trifolium dubium, Holcus lanatus, Daucus carota Achillea millefolium, Leucanthemum vulgare, Taraxacum officinale* and *Dactylis glomerata*. This vegetation community was found on Transects 1, 2 and 3, with the highest occurrence in Transect 1.

Among the reeds on the individual transects, in different frequencies and areas, reeds from the union of *Phragmition (Iridetum pseudacori* and *Phalaridetum arundinaceae)* were dominant. They often occurred in the form of a narrow strip along the riverbed. In Transect 1, the cause was *Helianthus tuberosus*, an invasive species for Poland, which grew along the escarpment in the form of single-species clusters. *Phalaris arundinacea, Urtica dioica* and *Reynoutria japonica* occurred singly within the communities *Callium aparine, Artemisia vulgaris* and *Solidago canadensis*.

The plant communities are dominated by an anthropogenic system with the participation of invasive species such as *Reynoutria japonica* and *R. sachalinensis*. This community forms a dense patch and, within its patch, there are no other species or just single species (Table 5). The height of individual specimens often reaches 2 m and even higher. This taxon dominates the entire area and is the main component of the willow riparian forest (*Salicetum albo-fragilis*). These species form single patches or occur together. In the second case, one of them is dominant.

**Table 3.** Floristic diversity in studied transects.

| Plant Names | T1 | | | T2 | | | T3 | T4 | | T5 | T6 | |
|---|---|---|---|---|---|---|---|---|---|---|---|---|
| | **A** | **B** | **C** | **A** | **B** | **C** | **A** | **A** | **C** | **C** | **A** | **C** |
| *Acer negundo* * L. | + | . | . | + | . | . | | + | + | . | . | + |
| *Acer platanoides* L. | . | . | . | . | . | . | . | + | + | . | + | + |
| *Acer pseudoplatanus* L. | | | | | | | | | | | + | + |
| *Achillea millefolium* L. | + | | | | + | | | | | | | |
| *Aegopodium podagraria* L. | | + | + | | + | | + | + | + | | + | + |
| *Agropyron repens* (L.) | + | | | | | | + | | | | | |
| *Agrostis gigantea* Roth | + | | | | + | | + | | | | | |
| *Alisma plantago-aquatica* L. | | | | | | | | | + | | | + |
| *Allium ursinum* L. | | | | | | | | | | | + | |
| *Alliaria petiolata* (M. Bieb.) Cavara et Grande | | | + | | + | | + | | | | + | |
| *Alnus glutinosa* (L.) | | + | + | | | | | | | | | + |
| *Anthriscus sylvestris* (L.) | | | | | | | | | | + | | |
| *Arctium lappa* L. | | | | | + | | | + | | | | |
| *Arrhenatherum elatius* (L.) | + | + | + | | + | | + | | | | | |
| *Artemisia vulgaris* L. | | | | + | | | + | | | | | |
| *Avena pratensis* L. | + | | | | | | + | | | | | |
| *Betula pendula* Roth. | | + | + | | | | | | | | | |
| *Caltha palustris* L. | | | | | | | | | + | | | + |
| *Campanula patula* L. | + | | | | + | | | + | | | | |
| *Carex flava* L. | | | | | | + | | | | | | |
| *Carex parviflora* Host | | | | | | + | | | | | | |
| *Cerasus avium* (L.) | | | | | | | | | + | | | |
| *Chelidonium majus* L. | | | | | | + | | | | | | |
| *Convolvulus arvensis* L. | | | | | | | | | | | + | |
| *Cornus alba* L. | | + | + | | | | | | | | | |
| *Cornus sanguinea* L. | | | | | | | | | + | | | |
| *Crataegus laevigata* (Poir.) | | | | | | | | + | + | | | |
| *Crataegus monogyna* Jacq | | | + | | | | | + | + | | | |
| *Crepis mollis* (Jacq.) | + | | | | | | | + | | | | |
| *Cynosurus cristatus* L. | + | | | | + | | + | | | | | |
| *Dactylis glomerata* L. | + | | | | + | | + | | | | | |
| *Equisetum arvense* L. | | + | + | | + | | | | | | | |
| *Euonymus europaeus* L. | | | | | | + | | | | | | |
| *Euphorbia esula* L. | + | | | | | | | | | | | |
| *Festuca pratensis* Huds. | + | | | | + | | | | | | | |
| *Fraxinus excelsior* L. | | | + | | | | + | + | + | | | |
| *Galeobdolon luteum* Huds. | | | + | | | | + | | | | + | |
| *Galeopsis tetrahit* L. | | | + | | | | + | | | | + | |
| *Galium aparine* L. | + | + | + | | | + | + | + | + | + | + | |
| *Galium mollugo* L. | + | | | | + | | + | | | | | |

**Table 4.** *Cont.*

| Plant Names | T1 | | | T2 | | | T3 | T4 | | T5 | T6 | |
|---|---|---|---|---|---|---|---|---|---|---|---|---|
| | **A** | **B** | **C** | **A** | **B** | **C** | **A** | **A** | **C** | **C** | **A** | **C** |
| *Galium verum* L. | | + | + | + | | | | | | | | |
| *Geum rivale* L. | | | | | | | | | + | | + | |
| *Glechoma hederacea* L. | | | | | | | | | | | + | |
| ***Helianthus tuberosus* L.** | + | | | | | | | | | | | |
| *Heracleum sphondylium* L. | | | + | | | | + | | | | | |
| *Holcus lanatus* L. | + | | | | + | | | | | | | |
| *Holcus mollis* L. | | | | | | | | | | | + | |
| *Humulus lupulus* L. | | + | + | | | + | | + | + | + | | |
| ***Impatiens glandulifera* Royle** | | | + | + | | | + | + | | + | | |
| ***Impatiens parviflora* DC.** | | | | | | + | | | | | | |
| *Iris pseudacorus* L. | | | + | | | | | | | | + | + |
| *Juglans regia* L. | | | | | | | | | | | | |
| *Juncus conglomeratus* L. | | | | | | | + | | + | | | + |
| *Lamium maculatum* L. | | | | | | | | | | | | |
| *Lemna minor* L. | | | | | | + | | | | | | |
| *Leontodon hispidus* L. | + | | | | + | | | | | | | |
| *Leucanthemum vulgare* Lam. | + | | | | + | | | | | | | |
| *Lotus corniculatus* L. | + | | | | + | | | | | | | |
| *Medicago falcata* L. | + | | | | + | | | | | | | |
| *Medicago lupulina* L. | + | | | | + | | | | | | | |
| *Medicago sativa* L. | + | | | | + | | | | | | | |
| *Mentha aquatica* L. | | | | | | + | | | | | | |
| *Myosotis palustris* (L.) L. em. Rchb. | | | | | | | | | | | | |
| *Nuphar lutea* Sm. | | | | | | | | | | | + | + |
| *Padus avium* Mill. | | | + | | | | | | | | | + |
| ***Padus serotina* (Ehrh.)** | | + | + | | | | | | | | | |
| ***Parthenocissus inserta* (A.Kern.) Fritsch** | | | | | | + | | | | | | + |
| *Phalaris arundinacea* L. | + | | + | | | | + | | + | | | |
| *Phragmites australis* (Cav.) | | | + | | | + | + | | + | + | | |
| *Pinus sylvestris* L. | | + | + | | | | | | | | | |
| *Plantago lanceolata* L. | + | | | | + | | | | | | | |
| *Populus nigra* L. | | + | + | | | | | | | + | + | + |
| *Populus tremula* L. | | + | + | | | | + | | | | | |
| *Potentilla erecta* (L.) | + | | | | + | | | | | | | |
| *Quercus robur* L. | | + | + | | | | | + | + | | + | + |
| ***Quercus rubra* L.** | | + | + | | | | | | | | | |
| *Ranunculus acris* L. | + | | | | | | | | | | | |
| ***Reynoutria japonica* Houtt.** | | + | + | + | | | | | + | + | | |
| ***Reynoutria sachalinensis* (F. Schmidt) Nakai** | + | + | + | | | | + | | | + | + | |
| *Rhamnus catharticus* L. | | + | + | | | + | + | | | | | |
| ***Robinia pseudacacia* L.** | | + | | | | | | | | | | |
| *Rosa canina* L. | | | | | | | | + | + | | | |

Table 5. *Cont.*

| Plant Names | T1 | | | T2 | | | T3 | T4 | | T5 | T6 | |
|---|---|---|---|---|---|---|---|---|---|---|---|---|
| | A | B | C | A | B | C | A | A | C | C | A | C |
| *Rubus idaeus* L. | | + | | | | | + | | | | | |
| *Sagittaria sagittifolia* L. | | | | | | + | + | | | | | + |
| *Salix alba* L. | | + | | + | | | + | + | + | + | + | + |
| *Salix caprea* L. | | + | | | | | + | + | | | | + |
| *Salix fragilis* L. | | + | | | | + | + | + | + | + | + | + |
| *Salix purpurea* L. | | | | + | | | | + | | + | | |
| *Salix triandra* L. | | | + | | | | | + | | + | | |
| *Salix viminalis* L. | | | | | | | | | | + | | |
| *Sambucus nigra* L. | | + | | | | | + | + | + | | + | + |
| *Sanguisorba officinalis* L. | + | | | | + | | | | | | | |
| *Schoenoplectus lacustris* (L.) | | | | | | | | | + | | | + |
| *Scutellaria galericulata* L. | | | | | | | + | + | + | + | | |
| **Solidago canadensis L.** | + | + | | | | | + | | | | + | |
| *Stellaria media* (L.) | | + | + | | | | + | | | | | |
| *Symphytum officinale* L. | | | | | + | | | | | | | |
| *Tanacetum vulgare* L. | + | | | + | + | | | | | | | |
| *Taraxacum officinale* coll. | | + | | | + | | | | | | | |
| *Tilia cordata* Mill. | | | | | | | | | | | | |
| *Tragopogon pratensis* L. | + | + | | | + | | | | | + | | |
| *Trifolium montanum* L. | + | | | | | | | | | | | |
| *Trifolium pratense* L. | + | | | | + | | | | | | | |
| *Trifolium repens* L. | + | | | | | | | | | | | |
| *Ulmus laevis* Pall. | | | | | | | + | | | + | | |
| *Urtica dioica* L. | + | + | | | | | + | + | + | + | + | |
| *Valeriana sambucifolia* J. C. Mikan | | | | | | | | | + | | | + |
| *Verbascum nigrum* L. | + | | | | | | | | | | | |
| *Veronica chamaedrys* L. | + | | | | | | | | | | | |
| *Vicia cracca* L. | + | | | | + | | | | | | | |
| *Vicia sepium* L. | + | | | | + | | | | | | | |
| *Vicia tetrasperma* (L.) | + | | | | + | | | | | | | |
| *Viola reichenbachiana* Boreau | | | | | | | | + | | | | |
| *Viscum album* L. | | | | | | | | | | + | | |
| Total | 40 | 29 | 31 | 7 | 30 | 15 | 32 | 23 | 26 | 18 | 22 | 21 |

Explanation: T: transect; A: inundation zone; B: dikes; C: oxbow lake; and bold font indicates an invasive species.

The *Impatiens glandulifera* association occurs on all the analyzed transects and in the entire Upper Odra Valley. It often coexists with *R. japonica* but, in comparison, it occupies a much smaller area. In the valley zone, anthropogenic communities are often formed on anthropogenic embankments, and their surface area is conditioned by the surface of an embankment or anthropogenic or industrial embankment. These include ruderal communities, mainly of the *Artemisitea vulgaris* class, represented mainly by the communities of *Artemisio-Tanacetetum vulgaris*, *Echio-Melilotetum* and *Dauco-Picridetum hieracioidis*. Communities of the *order Sisymbrietalia* and *Polygono-Chenopodietalia* also occur in small areas.

### 3.2.2. Forest Vegetation

From the forest communities in the study area, mainly the occurrence of *Salicetum albo-fragilis* and a community of shrub formations (*Salicetum purpurea* and *Salicetum triandro-viminalis*) were recorded. Shrub formation communities occupy a small area and do not play an important biocenotic role.

*Salicetum albo-fragilis* occurs on all analyzed surfaces and is rich in flora. Among tree species, it is often accompanied by *Populus nigra*, *Ulmus laevis* and *Salix alba* and from the shrubs *Salix viminalis*, *S. triandra*, *Sambucus nigra* and *S. purpurea*. The riparian undergrowth is dominated by *Urtica dioica* and *Galium aparine*, less often by *Scutellaria galericulata* and *Anthriscus sylvestris*. As already mentioned, the entire fleece is often covered with *Reynoutria japonica* and *R. sachalinensis*. These communities can be considered as remnants from before the period of river regulation, hence they have a strongly loosened tree stand and undergrowth changed by various anthropogenic activities, including fertilization of neighboring farmlands.

### 3.2.3. Flora Diversity

Transects differ in terms of floristic composition. The highest number of species was found in the floodplain for Transects 1 (40 species) and 3 (32) (Table 5). The lowest number of species was recorded in Transect 2 (floodplain), associated with the narrow strip and the dominance and aggregation of *S. canadensis*. Similar conditions prevailed on Transect 4. The floristic diversity is due to the presence of habitat mosaics and the species' wide range of ecological requirements. The most numerous species-rich fragments are fresh meadows of the *Arrhenatherion elatioris* assemblage developed on the dikes (Transect 1. Table 5).

## 4. Discussion

Before regulation works started in 18th century, the Odra River in the studied section was a meandering river. People settled in the area of Upper Odra as early as the Neolithic Period (7000 years BP) [37]. Over time, agriculture was developed, the first cities were founded and the Odra was used, among other purposes, to drive watermills. The first weirs for watermills on the Odra were built in the 12th century [32]. Since the Middle Ages, the riverbank revetments were also locally constructed to prevent the migration of the channel. Large-scale regulation of the Odra Channel for navigational purposes began in 1746 [33]. In the early phase, which lasted until the early 19th century, the main goal was to straighten the channel's course by cutting off meanders. In the section of the Odra River under study, the shortening of the river by cutting meanders was done between 1870 and 1922. The cutting off of the meanders has dramatically changed the character of the river. The sinuosity of the regulated riverbed drastically decreased, while its gradient and thus water energy increased [38]. This, in turn, resulted in erosion and a gradual channel incision by 2.5 m only since 1988 [14]. Therefore, it can be assumed that the groundwater level has also decreased, as shown by numerous studies [39–41], since the significant channel incision could not have left an impact on the riverine aquifer, soil moisture and flood-dependent ecosystems across the floodplain, but the exact relationship between groundwater dynamics and the functioning of riparian ecosystems requires further study, which may be particularly important in the context of prolonged droughts [41].

Changes in the course of the channel in the meandering section were also observed during the analyzed period. Lateral migration of the channel of a meandering river is here a natural process, as is the spontaneous cutting of the meander neck during floods. This particular, semi-natural section of the Odra Channel has been protected under the Natura 2000 program since 2009. In the section in question, one of the meanders in Transect 4 was naturally cut off in 1966 and the other, in Transect 6, in 1997. The oxbows that were created in the process established new conditions for the plants because running water was replaced by stagnant water. The meander in Transect 4 was naturally cut off in 1966, but is still connected to a nearby pond, and fresh water from the pond overflows into the oxbow lake, so the reservoir is not overgrown (Figure 4C). In contrast, the meander in Transect 6, which was cut off in 1997, has no fresh water inflow and is slowly becoming overgrown (Figure 4A). The oxbow in Transect 2 is also supplied with water by a small water course. The supply of fresh water seems to be sufficient to prevent overgrowth of the reservoir (Figure 4B). The most significant changes in land cover between 1910 and 2022 can be seen in the loss of arable land by 45.7%, an increase in water-covered areas by 15.2%, an increase

in forested areas by 14.3% and an increase in grassland areas by 10.9%. Residential areas also increased by 5.4% (Table 2). All these changes are the result of changes in land use. After the river section was channelized, the valley floor was intensively used for agriculture (Figure 3A,B), but with the passage of time and the development of the economy, the demand for construction materials such as gravel and sand has increased. The Odra's floodplains offer easily available supplies of these materials. Gravel pits fill with water after mining, thereby increasing the total area of water bodies on the floodplain (Figure 3B,C).

Riparian forest, as a flood-dependent ecosystem, is potentially an essential element of the riverine landscape [42]. However, the development of agriculture on fertile floodplains and deforestation in accordance with misconceived flood protections have led to the defragmentation of riparian forests and, in places, to their complete disappearance [30,43–45]. This problem also applies to the section of the Upper Odra Valley discussed in this article, where areas of potential riparian forests have been taken up for crops and meadows. At the same time, an increase in the area of riparian forest is observed in the regulated section as a result of an increase in the area of stagnant water. However, contrary to natural conditions, these forests do not overgrow areas adjacent to the riverbed, but are the result of succession in places where stagnant water has appeared due to human activity.

An analysis of cartographic materials (1910–2022) confirms that the changes that occurred during this period were not drastic, as in the case of the previous regulation of the river channel in 1882 (Figure 1). Therefore, it can be assumed that the final formation of the actual vegetation in the zone of the river influence (the floodplain) was impacted by the development of agriculture and the straightening of the river channel. These two factors often interacted together. The third factor that contributed to the mosaic development of vegetation in the regulated section was the extraction of gravel in the cut-off bends. In the natural section, agricultural development directly along the river played a major role in this process.

These changes alter the biodiversity of the floodplain, e.g., fragmentation of the riparian forest forming a mosaic of wooded patches in agricultural land along the entire studied section. Such a situation is common in the floodplains of European rivers [25].

Our study indicates that the degree of transformation of the river channel influences the diversity of vegetation species but, paradoxically, like Dufour et al. [28], we found that greater biodiversity characterizes the riparian zone of the regulated section. At the same time, the occurrence of species there that form a species-rich fresh meadow is not strictly related to the presence of the river. The aforementioned diversity of species is related to the presence of artificial dikes built of clayey and even gravelly soil material. This increases the water capacity of the soil and contributes to the formation of microhabitats and the encroachment of species with different ecological requirements [16,17,19,20]. An additional cause is maintenance work on the embankments involving mowing grasses and removing tree seedlings. Habitats typical for riparian areas have a poorer species composition, but their value should be seen in sustaining the functioning of natural ecosystems. Therefore, even if the number of species is more limited in riparian forests, they are valuable because of their nativeness and close relationship with the biotope. A return to conditions prior to regulation and human influence is an impossible ideal that those undertaking revitalization efforts are striving for, because the current state is the result of a whole range of human activities that have lasted for millennia [26,27]. The results relate to studies conducted in other regions of Poland indicating the impact of anthropopressure on species diversity resulting from the formation of habitat complexes [20,29,46]. In the regulated section of the Odra, an initial ecosystem was highly disturbed by meander cut offs and mining activity at the floodplain. After a dynamic period of change on the floodplain relief, a new equilibrium has been established and conditions for vegetation are now more stable, especially for the areas on the outer side of the dikes. Frequent disturbances in the form of floods occur only in the narrow interembankment zone where the highest number of species was recorded. Studying species diversity along the meandering stretch of the Odra, we observed that a more near-natural riparian system is not automatically richer in species.

Similar relationships have been observed on German rivers, including the lower Odra course [29].

The small number of species is also influenced by invasive vegetation. The distribution of invasive alien species in the surveyed sections of the valley is irregular. Areas of uncultivated land along the river are massively overgrown with *R. japonica* and *R. sachalinensis*. Their range often covers areas about 300 m from the river banks, most often uncultivated areas. They also develop within willow and poplar riparian forests and are often the main species of these forest ecosystems. It should be noted that these are open areas, where solar radiation reaches the ground at 100%. These are species of very high vitality and within them other species occur rarely. Hence the ecosytems of the Odra valleys in the meandering section are floristically poor.

Invasive species threaten native ecosystems around the world [18,22,45,47]. Rivers as ecological corridors promote the propagation of invasive species and riparian areas are prone to be colonized by alien plant species [45,48,49]. In Poland, not only riparian forests but also other habitats like alluvial meadows [50] or mountain grasslands [51] are threatened by this phenomenon.

*R. japonica* and *R. sachalinensis* were found on the banks of the surveyed section of the Odra River, and are species from East Asia, growing there along gullies and streams in the mountains from Sakhalin to Japan. In Europe, it has been cultivated since the second half of the 19th century as an ornamental and melliferous perennial. Quickly going wild from cultivation, invasive and much expansive. The first synanthropic sites were recorded in Poland in the late 19th century. It continues its expansion. In Poland, it is more common in the south of the country, growing in riparian areas and in ruderal habitats [48,52].

The spread of invasive plant species is a significant problem for native riparian ecosystems especially in the context of climate change. This is a major challenge facing management policy makers to protect future ecosystems from these combined threats [22,23].

## 5. Conclusions

Biocenotic systems are among the dynamically changing elements of the geosystem and some are naturally replaced by others. Changes in ecosystems are caused by both natural and anthropogenic factors and, in addition, anthropogenic effects can affect the environment directly or indirectly. Anthropogenic effects (more than 100 years after channel regulation) have been found in the analyzed section of the upper Odra River.

The floristic diversity on different transects shows the diversity of Odra floodplain habitats. Paradoxically, the greatest diversity of plant species was found along the channelized section of the river with dikes built parallel to the riverbed.

Centuries of human activity to regulate the riverbeds for flood control and agricultural use of fertile soils on the floodplains have led to changes in the geometry of meandering river channels and associated ecological systems. The most drastic changes in the form of regulation of the European river (channelization) were recorded in the 18/19 w period.

The surveys conducted indicate the following in various sections of the river:

1. As a result of engineering works and straightening the river channel, the mosaic of wetlands and natural floodplains has been completely transformed, including willow meadows.
2. The areas associated with the dikes are rich in plant species and form fresh grasslands that are not dependent on the presence of the river for their development.
3. Most of the floodplain is colonized by geographically alien species. The areas they occupy are poor in terms of floristic diversity. A contemporary threat to the ecosystems of the Odra Valley, also in this section, is the mass occurrence of alien species such as *R. japonica, R. sachalinensis* and *I. glandulifera*.
4. In spite of increased anthropopressure, the riparian forest *Salicetum-fragilis* has been preserved in this section of the Odra River almost unchanged in terms of species composition. This complex is also one of the important components of the forest landscape in this area.

**Author Contributions:** Conceptualization, A.C. and O.R.; methodology, A.C. and O.R.; validation, A.C. and O.R.; formal analysis, A.C. and O.R.; investigation, A.C. and O.R.; resources, A.C. and O.R.; data curation, A.C., B.S. and O.R.; writing—original draft preparation, A.C. and O.R.; writing—review and editing, A.C., O.R. and B.S.; visualization, A.C. and B.S. All authors have read and agreed to the published version of the manuscript.

**Funding:** This research received no external funding.

**Data Availability Statement:** Data will be made available directly by the authors upon request.

**Conflicts of Interest:** The authors declare no conflict of interest.

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
