# Peer review of "The Natural Consequences of Land Use Change on Transformation and Vegetation Development in the Upper Odra Floodplain"

_water, doi:10.3390/w15193493_

Round 1

Reviewer 1 Report

It seems to me a very current issue in which many European countries work in order to obtain answers to the changes in land use in the stretches of rivers that have suffered a series of anthropic imbalances, generally throughout the twentieth and twenty-first centuries.

I think the segregation of a section of the Odra River is good, and the selection of 6 transects to study its evolution, but it could have been more ambitious, choose more sections and compare this same study that the authors propose to other sections downstream and upstream with the aim of drawing conclusions about longitudinal connectivity,  A very important part of any river restoration process and the evolution of riparian vegetation along the course of the river

The process of the evolution of the affected areas in each of the transects are very well explained according to the objective of the work and the results obtained from the variations of the occupied surfaces are very important for the management of the transnational basin of the river.

Only a hydraulic modelling of the section would be missing to check the behaviour of flows generated by extreme rainfall, which would allow a technical discussion on current land uses and flood behaviour. Maybe you could to mention somenthing about this

 Change the reference of century  because I understand is 20nd century

It strikes me that the areas associated with the dams are rich in plant species and form fresh grass, lands that do not depend on the presence of the river for their development. Could you explian more about this?

Author Response

Response in an attachment

Reviewer 2 Report

The Authors describe the impact on the vegetation in the upper Odra river area, induced by human activity, starting from 1910 to the present day. In particolar, Odra river shows a different nature, and it can be briefly divided into channalized and naturally meandering areas.

The paper deals with an interesting and current topic, although very vast. The Authors clearly describe the situation of the study area, its evolution over time, the effects of human activity on vegetation diversity.

The Discussion could be improved, especially by linking it more effectively to the Results. It would also be interesting for the Authors to better discuss the applicability of the definition of Indirect Disturbance Hypothesis to the case under study.

Consequently, the Conclusions could be improved.

The quality of English Language is good.

Author Response

Response in an attachment
